# Root Traits and Biomechanical Properties of Three Tropical Pioneer Tree Species for Forest Restoration in Landslide Areas

**Jung-Tai Lee** [1,*], **Ming-Yang Chu** [1], **Yu-Syuan Lin** [1], **Kuan-Ning Kung** [2], **Wen-Chi Lin** [3] and **Ming-Jen Lee** [1]

[1] Department of Forestry and Natural Resources, Graduate School of Agricultural Sciences, National Chiayi University, Chiayi 60004, Taiwan; s1060103@mail.ncyu.edu.tw (M.-Y.C.); s1070176@mail.ncyu.edu.tw (Y.-S.L.); mjlee@mail.ncyu.edu.tw (M.-J.L.)

[2] Liouguei Research Center, Taiwan Forest Research Institute, Kaohsiung 84443, Taiwan; kkn@tfri.gov.tw

[3] Chiayi Forest District Office, Taiwan Forestry Bureau, Chiayi 60049, Taiwan; jmf55121343@yahoo.com.tw

[*] Correspondence: jtlee@mail.ncyu.edu.tw; Tel.: +886-5-271-7482

**Abstract:** Frequent earthquakes, monsoon torrential rains and typhoons cause severe landslides and soil erosion in Taiwan. *Hibiscus taiwanensis*, *Macaranga tanarius*, and *Mallotus paniculatus* are major pioneer tree species appearing on landslide-scarred areas. Thus, these species can be used to restore the self-sustaining native vegetation on forest landslides, to control erosion, and to stabilize slope. However, their growth performance, root traits and biomechanical properties have not been well characterized. In this study, root system and root traits were investigated using the excavation method, and biomechanical tests were performed to determine the uprooting resistance, root tensile strength and Young's modulus of 1-year-old *Hibiscus taiwanensis*, *Macaranga tanarius*, and *Mallotus paniculatus* seedlings. The results reveal that relative to *H. taiwanensis*, *M. tanarius* and *M. paniculatus* seedlings had significantly larger root collar diameter, longer taproot length, higher root biomass, higher root density, higher root length density, heavier root mass, larger external root surface area, higher root tissue density, larger root volume, longer total root length, and a higher root tip number. Additionally, the height of *M. paniculatus* seedlings was significantly higher than those of *H. taiwanensis* and *M. tanarius*. Furthermore, the uprooting resistance and root tensile strength of *M. paniculatus* seedlings was significantly higher than those of *H. taiwanensis* and *M. tanarius*. Young's modulus of *M. paniculatus* and *M. tanarius* seedlings was also significantly higher than that of *H. taiwanensis*. These growth characteristics and biomechanical properties demonstrate *M. paniculatus* and *M. tanarius* are superior than *H. taiwanensis*, considering growth performance, root anchorage capability, tensile strength and Young's modulus. Taken as a whole, the rank order for species selection of these pioneer species for reforestation comes as: *M. paniculatus* > *M. tanarius* > *H. taiwanensis*. These results, along with knowledge on vegetation dynamics following landslides, allow us to better evaluate the effect of selective removal management of pioneer species on the resilience and sustainability of landslides.

**Keywords:** landslides; pioneer tree species; root architecture; root traits; root biomechanical properties; forest restoration

---

## 1. Introduction

Taiwan, a mountainous island, is one of the most vulnerable places to natural hazards on the Earth [1]. Typhoons and earthquakes are the most serious and destructive hazards that cause many landslide disasters in forestland areas in Taiwan [2]. Landslide restoration and soil stabilization are major issues in forest ecosystem management. Drought, instability, poor soil fertility and steep slopes

are typical characteristics of forest landslides. Pioneer tree species play an important role in landslide restoration and soil conservation [3]. Successional reclamation can enhance natural successional processes for rehabilitation of drastically disturbed sites [4]. Soil bioengineering and successional reclamation can be applied to restore of self-sustaining native forest ecosystem [5]. In Hong Kong and Taiwan, soil bioengineering has been employed to enhance the stability of slopes [6,7]. Chou et al. [8] conducted a 3-year investigation of the vegetation dynamics of early succession in a 6.67 ha landslide area located at Shanping, southern Taiwan (216429.43 E, 2541177.10 N, TWD 97). They demonstrated the landslide area was primarily dominated by native pioneer trees, such as *Hibiscus taiwanensis*, *Macaranga tanarius*, and *Mallotus paniculatus*. These species are endemic pioneer shade-intolerant tree species, distributed from low altitudes up to 1800 m in landslide scars [9,10] and are gradually succeeded by tree species, such as *Cinnamomum micranthum*, *Machilus thunbergii*, *Castanopsis formosana*, *Castanopsis indica* and *Cyclobalanopsis glauca* [11]. *M. tanarius* was also listed as critically endangered on the IUCN Red List of Threatened Species [12]. Previous studies have indicated that these three tree species have high potential for landslide restoration and soil bioengineering [13,14].

Tree root system architecture and biomechanical properties have great influence on soil slope stability. Root system architecture specifies the pattern and spatial distribution of the root systems [15]. Patterns of plant root system architecture have been classified into five types, i.e., H- (horizontal), R- (right), VH- (vertical and horizontal), V- (vertical) and M- (massive) type [16]. Root anchorage capability and soil reinforcement force are important factors regarding slope stability. Saifuddin et al. [17] showed that *Leucaena leucocephala* has higher root biomass, root length and root tensile strength, indicating a higher contribution to anchorage capability than *Peltophorum pterocarpum*. Stokes et al. [18] indicated that plant root pullout resistance is significantly affected by several morphological traits, such as rooting depth, the proportion of fine roots and root architecture. There have been few studies conducted on root traits and biomechanical properties of *H. taiwanensis*, *M. tanarius*, and *M. paniculatus*. Yen [16] showed that root system architecture of these three species belonged to VH-type. Lin et al. [19] indicated that the shear strength increment of *H. taiwanensi* ranged between 19.9 and 36.3 kPa. Avani et al. [20] showed that *M. tanarius* distributed about 50% of its roots in the first 10 cm soil layer. Lateh et al. [21] indicated that root tensile strength of *M. tanarius* was significantly higher at 25 cm compared to 1 m from the tree stem base. However, the root traits and biomechanical properties of these three species have not yet been fully investigated. This study aimed to (1) investigate the root traits and biomechanical properties of these species and (2) compare these differences among species to apply in soil bioengineering projects for slope stabilization and landslide mitigation.

## 2. Materials and Methods

### 2.1. Seed Collection

Mature elite trees of *H. taiwanensis*, *M. tanarius*, and *M. paniculatus* were selected from the landslide forest stand located at Dapu Township, Chiayi County, Taiwan (120°34′59″ E, 23°19′51″ N) in May 2017. Capsules were collected from the upper crown of mature trees, in June, October and December, 2017, respectively. Capsules were sun-dried in trays for seed release. Seeds were extracted, cleaned, put in polyethylene bags, and stored at room temperature.

### 2.2. Seedling Preparation

In January 2018, seeds of three species were surface cleaned with tap water, sterilized 2 times with 10% sodium hypochlorite solution for 10 min and rinsed with sterile water 3 times, and then germinated in autoclaved peat moss, perlite and vermiculite mixtures (1:1:1, v/v). Wooden containers (*l×w×h*, 30 cm × 30 cm × 100 cm) were used for transplanting. Before transplanting, the containers were filled with sandy silt loam soils (consisting of 72.4% sand, 20.3% silt and 7.3% clay; specific gravity 2.66; bulk density 1.35 g cm$^{-3}$, porosity 43%) collected from the same landslide forest stand. The chemical properties of soil are given in Table 1. When 3-month-old seedlings attained a mean

height of 10.2 ± 3.3 cm and root collar diameter of 2.3 ± 0.15 mm in April 2018, thirty-two seedlings of each species were transplanted into the containers separately, randomly arranged in the forest nursery of National Chiayi University (23°28′12″N, 120°29′22.3″E) under ambient conditions, and watered every morning. Containers with seedlings were rotated weekly to minimize differences in illumination among seedlings. Meanwhile, the azimuth of each container was not changed during rotation.

**Table 1.** Chemical properties of dry soil in the present study.

| Properties | Soil |
|---|---|
| pH (water) | 7.98 |
| Conductivity (dS m$^{-1}$) | 0.13 |
| Organic carbon (g kg$^{-1}$) | 0.07 |
| Total nitrogen (%) | 0.03 |
| Phosphorus (mg kg$^{-1}$) | 21 |
| Potassium (mg kg$^{-1}$) | 152 |
| Calcium (mg kg$^{-1}$) | 3491 |
| Magnesium (mg kg$^{-1}$) | 320 |
| Zn (ppm) | 4.3 |
| Mn (ppm) | 163 |
| Fe (ppm) | 343 |
| Cu (ppm) | 1.8 |
| Cd (ppm) | 0.01 |
| Cr (ppm) | 0.74 |
| Ni (ppm) | 1.5 |
| Pb (ppm) | 4.4 |

### 2.3. Growth Performance and Root System Architecture Investigation

A pilot study revealed the containers had enough room to allow the root growth during the study period, and the size and distribution of roots were not constrained by the container. Twelve months after transplanting, fourteen seedlings of each species were randomly chosen for growth performance, root system architecture and root trait investigations. Before excavation, seedling height (H) and root collar diameter (RCD) were recorded. Root systems were carefully extracted from the soil by flushing with water to prevent root damage. The rooting depth was measured and root numbers were counted. Root area ratio (RAR) values were obtained by collecting all roots (1–10 mm diameter) from each 10 cm soil layer using a hand excavation method [22], and computed as total root area $A_r$/area of soil profile $A$ [23]. Root characteristics were investigated and recorded. Images of root systems were taken and stored for further investigation of root architecture. Root traits, e.g., external root surface area (RSA), root tip number (RT) and total root length (TRL) were measured using WinRHIZO*Pro* Image Analysis System (Regent Instruments, Quebec, Canada) [24], whereas root volume was measured using water displacement to avoid overestimation [25,26]. Leaf, stem, and root dry biomass were measured by drying in a hot air oven at 70 °C for 48 h. These measurements were used to calculate root traits, such as, root density (RD, root dry mass/soil volume, kg m$^{-3}$), root length density (RLD, root length/soil volume, km m$^{-3}$), root mass (RM, root dry mass/soil volume, g m$^{-3}$), root surface area (RSA, cm$^2$), root tissue density (RTD, root dry mass/root volume, g cm$^{-3}$) and specific root length (SRL, root length/root dry mass, m g$^{-1}$) [27,28]. Live roots were also collected from sampled seedlings and prepared for subsequent root tensile strength and Young's modulus tests.

### 2.4. Vertical Pullout Test

Meanwhile, fourteen seedlings of each species were randomly sampled for pullout testing. The soil material was sandy silt loam, with an average dry weight of 15.3 kN m$^{-3}$, and moisture content of 28 ± 3% measured in the center of soil body at 30-cm depth using Rixen M-700 soil moisture meter (Rixen Tech Co. Ltd., Taipei, Taiwan). This condition was selected for providing consistent soil moisture parameters in all tests and avoiding wet soil effects on pullout resistance [29]. To evaluate pullout resistance, *in situ*

pullout tests were carried out with fourteen seedlings per species. Before each pullout test, seedling height and root collar diameter were measured with a ruler and digital caliper. The plant stem was cut off 15 cm above the stem base and debarked to prevent slippage of the pulling fixture. Pullout tests were carried out using a pullout machine (USPA-003, U-Soft Tech Co., Taipei, Taiwan) equipped with 5T load cell (Kyowa, Tokyo, Japan) [30]. The displacement was measured with a displacement transducer (TML DP-2000F, Tokyo Measuring Instruments Laboratory, Tokyo, Japan). Afterwards, the pullout machine was fastened to the pulling unit and a vertical traction force was applied automatically with a constant rate of 2 mm min$^{-1}$ until the pullout resistance force dropped noticeably. The maximum vertical resisting force ($F_{max}$, N) was recorded for further analysis.

### 2.5. Root Tensile and Young's Modulus Tests

After seedling root system excavation, the roots were cleaned carefully with tap water, and live roots in soil depth of about 30 cm were randomly sampled from seedlings of three species, respectively [31]. The root samples were categorized into four diameter classes (0–1, 1–2, 2–5, and 5–10 mm) [32]. Any damaged root segments were discarded. Sixty root segments were randomly sampled from seedlings of each species. Root samples were then cut into 60 mm in length, cleaned with tap water, put into zipper bags and stored at 4 °C on moist filter paper [33,34]. Biomechanical tests were performed within 48 h of collection. Tensile tests were performed in the laboratory using a tensile testing machine (USPT-003, U-Soft Tech, Taipei, Taiwan) fitted with a LCN-A load cell (Kyowa, Tokyo, Japan, maximum force 500 N, resolution 0.1 N) and a DP-2000F displacement transducer (TML, Tokyo, Japan). Data for tensile force and displacement were acquired and compiled with an automatic data acquisition system (U-Soft DA-8, U-Soft Tech, Taipei, Taiwan). Before testing, root diameter at the middle of segment was measured with digital caliper. The cross sectional area of root derived from root diameter was used to calculate tensile strength and Young's modulus. Root segments of 10 mm were then fastened to two clamps with screw grips. Subsequently, the root segments were subjected to a tensile test at a constant axial extension rate of 4.7 mm min$^{-1}$ till fracture. Successful test data resulting in fractures at the midpoint of the segment between the two clamps were recorded and analyzed. For each species, 60 root segments were tested in the middle section: 29 root segments of *H. taiwanensis* seedlings, 28 root segments of *M. tanarius* seedlings and 32 root segments of *M. paniculatus* seedlings. The tensile strength ($T_{si}$, MPa) was computed using the following formula [35–37]:

$$T_{si} = \frac{4F_{max}}{\pi d_i{}^2} \tag{1}$$

where $F_{max}$ is the maximum tensile force at fracture (N), and $d_i$ is the root segment diameter (mm) measured at the midpoint.

Root tensile strength is influenced as much by species as by root diameter. The relationship between root tensile strength ($T_s$) and diameter ($d$) can be expressed in the form of a simple power equation [36,38]:

$$T_s = \alpha \cdot d^{-\beta} \tag{2}$$

where $\alpha$ and $\beta$ are empirical constants depending on species.

Young's modulus ($E_r$) of each root sample was derived from the initial gradient of the tensile stress–strain curve within the initial linear elastic region using the following equation [33,39]:

$$E_r = \frac{F/A_0}{\Delta L/L_0} \tag{3}$$

where *F* is the maximum stress force (N), $A_0$ is the original cross sectional area, $\Delta L$ is the change in root length (mm), $L_0$ is the initial length of the root segment (mm).

*2.6. Data Analysis and Statistics*

Variation in growth parameters, root morphological traits and biomechanical properties among species were analyzed with one-way analysis of variance (one-way ANOVA) and means compared using Tukey's HSD post hoc test ($p < 0.05$) in IBM SPSS Version 22.0 statistical software package (SPSS Inc., Chicago, IL, USA). Descriptive statistics of root collar diameter, root biomass and shoot biomass were standardized by $[(x_i - \bar{x})/SE]$, analyzed with multicollinearity tests and multiple linear regression analysis using SPSS. Multicollinearity diagnostic tests were executed using Multiple Collinearity Regression by SPSS. Multiple regressions were carried out using Multiple Regression Analysis in SPSS to examine relationships between pullout resistance and root traits among species. Microsoft Excel Regression analysis (Excel 2013) was used to analyze the relationships between root tensile resistance, tensile strength, Young's modulus, and root diameter among species.

## 3. Results

*3.1. Root System Architecture*

*M. paniculatus* and *M. tanarius* seedlings developed deeper and more profuse root systems than *H. taiwanensis* seedlings (Figure 1). The root systems for *H. taiwanensis*, *M. tanarius* and *M. paniculatus* were categorized as VH- (vertical and horizontal) type according to the root system classification suggested by Yen [16]. *H. taiwanensis* exhibited shallower roots and shorter taproot systems, while *M. tanarius* and *M. paniculatus* possessed deeper taproots and more profuse lateral roots. *H. taiwanensis* seedlings developed taproots to 30 cm deep in soil, with most of the root biomass distributed in the upper 40 cm of the soil (Figure 1a). In contrast, *M. tanarius* seedlings grew taproots to 60 cm deep, with most of the root dispersed in the upper 60 cm soil layer (Figure 1b), whereas the taproot of *M. paniculatus* seedlings extended down to 70 cm depth of soil, with most of the root scattered in the upper 80 cm of the soil (Figure 1c). RAR (root area ratio, total root area $A_r$/area of soil profile $A$) was estimated for fourteen seedlings each of the three different species. RAR analysis showed high variation within species and significant differences among species in soil depth 0–10 cm (Table 2). RAR distribution showed that most of the root system spread in the upper 30 cm soil layer and root density decreased with soil depth (Figure 2). *M. tanarius* and *M. paniculatus* seedlings had significantly higher RAR than that of *H. taiwanensis* at 0–10 cm soil layer.

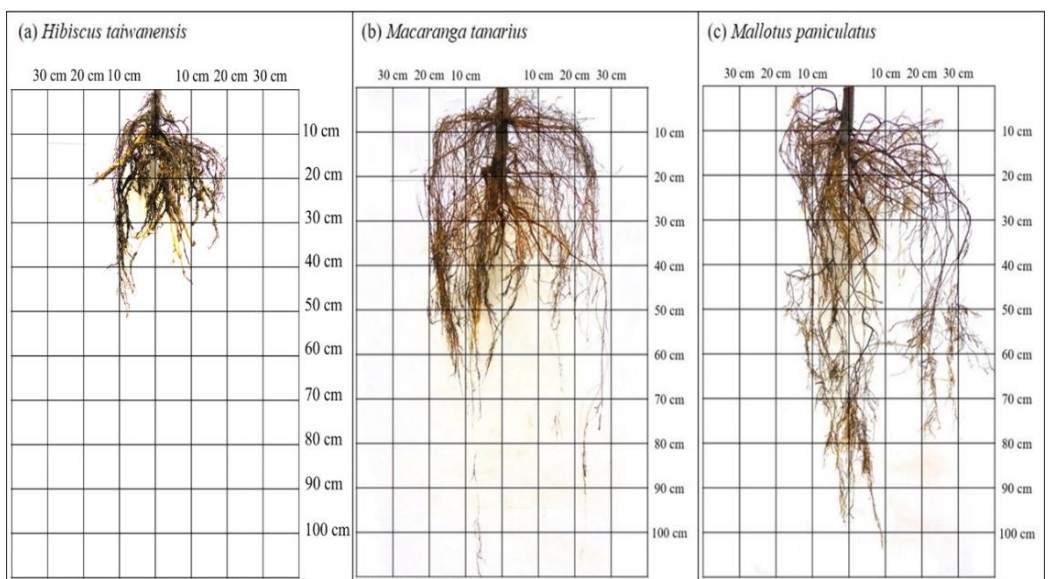

**Figure 1.** Typical root system architecture of 1-year-old *Hibiscus taiwanensis* (**a**), *Macaranga tanarius* (**b**) and *Mallotus paniculatus* (**c**) seedlings.

**Table 2.** Means ± standard deviations of root area ratio at different soil depths for the three species studied.

| Species | Root Area Ratio (%) | | | | | | |
|---|---|---|---|---|---|---|---|
| | 0–10 cm | 10–20 cm | 20–30 cm | 30–40 cm | 40–50 cm | 50–60 cm | 60–70 cm |
| *H. t.* | 0.31 ± 0.04[b] | 0.70 ± 0.11[a] | 0.44 ± 0.14[a] | 0.13 ± 0.071[a] | 0.04±0.037[a] | - | - |
| *M. t.* | 0.68 ± 0.05[a] | 0.81 ± 0.13[a] | 0.61 ± 0.09[a] | 0.26±0.055[a] | 0.1 ± 0.028[a] | 0.06 ± 0.017[b] | 0.02 ± 0.006[a] |
| *M. p.* | 0.67 ± 0.08[a] | 1.10 ± 0.15[a] | 0.49 ± 0.06[a] | 0.22±0.028[a] | 0.12 ± 0.02[a] | 0.12 ± 0.28[a] | 0.06 ± 0.03[a] |

*H. t.*, *Hibiscus taiwanensis*; *M. t.*, *Macaranga tanarius*; *M. p.*, *Mallotus paniculatus*. Different superscript letters in the same column indicate significant differences (ANOVA and Tukey's HSD (honestly significant difference) post hoc test) among species. $N = 14$. Level of significance is: $p < 0.05$.

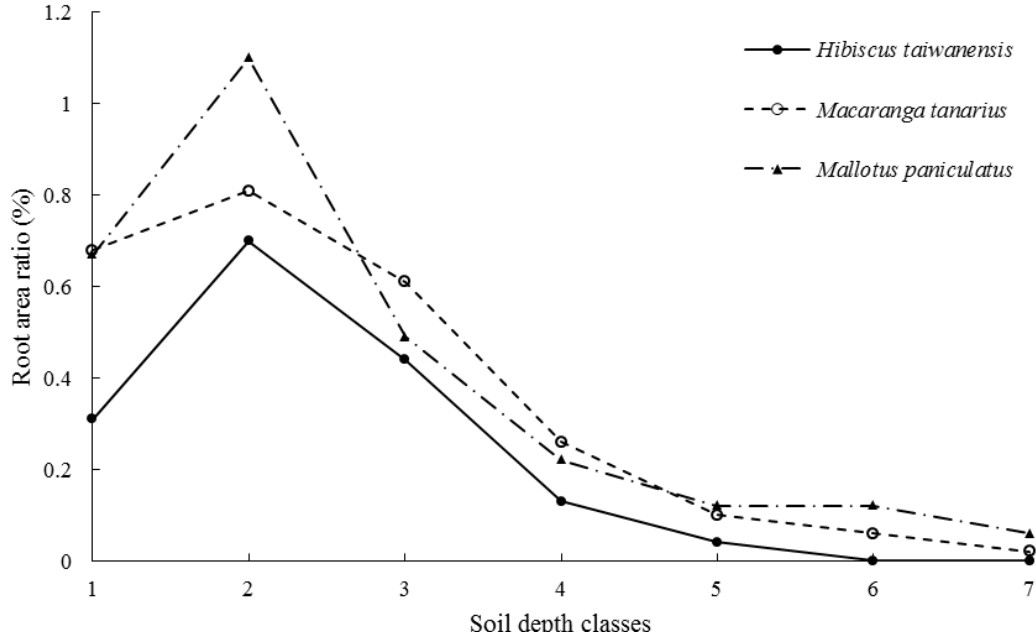

**Figure 2.** Root area ratio distribution with soil depth classes for the three species studied. Root area ratio (RAR = total root area $A_r$/area of soil profile $A$). Class 1 (0–10 cm), Class 2 (10–20 cm), Class 3 (20–30 cm), Class 4 (30–40 cm), Class 5 (40–50 cm), Class 6 (50–60 cm), Class 7 (60–70 cm).

*3.2. Seedling Growth Performance*

Results showed great variability in growth performance and root traits among species (Table 3). On average, plant height of *M. paniculatus* (161.1 ± 13.9 cm) was significantly higher than that of *H. taiwanensis* (142.6 ± 10.2 cm) and *M. tanarius* (138.1 ± 15.3 cm). Root collar diameter differed significantly among species and was highest for *M. tanarius* (25.0 ± 1.9 mm) and *M. paniculatus* (24.1 ± 2.3 mm) and lowest for *H. taiwanensis* (16.7 ± 0.9 mm). Taproot length was also significantly different among species and was highest for *M. tanarius* (83.1 ± 18.1 cm) and *M. paniculatus* (78.4 ± 15.2 cm) and lowest for *H. taiwanensis* (44.3 ± 10.1 mm). Meanwhile, the number of root tips of *M. tanarius* (10,780 ± 1944) and *M. paniculatus* (7905 ± 1811) were significantly higher than that of *H. taiwanensis* (5391 ± 2426). Total root length was significantly longer for *M. tanarius* (7474.2 ± 137.9 cm) and *M. paniculatus* (6447 ± 133.0 cm) than for *H. taiwanensis* (3950.7 ± 166.2 cm). Root biomass was significantly different among species and was highest for *M. paniculatus* (104.8 ± 11.4 g) and *M. tanarius* (96.3 ± 9.4 g) and lowest for *H. taiwanensis* (57.6 ± 16.4 g). Shoot biomass also differed significantly among species, and was highest for *M. paniculatus* (146.0 ± 32.4 g) and lowest for *H. taiwanensis* (59.5 ± 11.8 g) and *M. tanarius* (58.3 ± 6.9 g). Altogether, *M. paniculatus* and *M. tanarius* seedlings showed significantly higher growth performance than *H. taiwanensis* seedlings in this study.

**Table 3.** Means ± standard deviations of growth performance for the three species studied and F-values for one-way ANOVA.

| Growth Parameters | *H. taiwanensis* | *M. tanarius* | *M. paniculatus* | ANOVA |
|---|---|---|---|---|
| H (cm) | 142.6 ±10.2[b] | 138.1 ± 15.3[b] | 161.1 ± 13.9[a] | 4.388[*] |
| RCD (mm) | 16.7 ± 0.9[b] | 25.0 ± 1.9[a] | 24.1 ± 2.3[a] | 44.892[***] |
| TL (cm) | 44.3 ± 10.1[b] | 83.1 ± 18.1[a] | 78.4 ± 15.2[a] | 9.362[**] |
| RT | 5391 ± 2426[b] | 10780 ± 1944[a] | 7905 ± 1811[a] | 18.614[***] |
| TRL (cm) | 3950.7 ± 166.2[b] | 7474.2 ± 137.9[a] | 6447.4 ± 133.0[a] | 18.796[***] |
| RB (g) | 57.6 ± 16.4[b] | 96.3 ± 9.4[a] | 104.8 ± 11.4[a] | 44.846[***] |
| SB (g) | 59.5 ± 11.8[b] | 58.3 ± 6.9[b] | 146.0 ± 32.4[a] | 17.621[***] |

H, shoot height; RCD, root collar diameter; TL, taproot length; RT, root tips; TRL, total root length; RB, root biomass; SB, shoot biomass. Different superscript letters in the same row indicate significant differences (Tukey's HSD post hoc test) among species. $N = 14$. Levels of significance are: [*]$p < 0.05$; [**]$p < 0.01$; [***]$p < 0.001$.

### 3.3. Root Traits

The statistical analysis of root characteristics from WinRHIZO*Pro* showed that root traits were significantly different among species, except for root volume and specific root length (Table 4). Mean root density differed significantly among species and was highest for *M. paniculatus* (1.94 ± 0.21 kg m$^{-3}$) and *M. tanarius* (1.78 ± 0.17 kg m$^{-3}$) and lowest for *H. taiwanensis* (1.07 ± 0.30 kg m$^{-3}$). Root length density was significantly different among species and was highest for *M. tanarius* (1.38 ± 0.26 km m$^{-3}$) and *M. paniculatus* (1.19 ± 0.21 kg m$^{-3}$) and lowest for *H. taiwanensis* (0.73 ± 0.31 km m$^{-3}$). Root mass differed significantly among species and was highest for *M. paniculatus* (1939.8 ± 210.8 g m$^{-3}$) and *M. tanarius* (1782.4 ± 174.6 g m$^{-3}$) and lowest for *H. taiwanensis* (1067.4 ± 302.9 g m$^{-3}$). Root surface area was also significantly different among species and was highest for *M. tanarius* (5391.4 ± 786.8 cm$^2$) and then *M. paniculatus* (5277.4 ± 325.4 cm$^2$) and lowest for *H. taiwanensis* (2995.3 ± 959.4 cm$^2$). Root tissue density also differed remarkably among species and was highest for *M. paniculatus* (0.29 ± 0.02 g cm$^{-3}$) and *M. tanarius* (0.29 ± 0.04 g cm$^{-3}$) and lowest for *H. taiwanensis* (0.15 ± 0.03 g cm$^{-3}$). Overall, the root growth characteristics of *M. paniculatus* and *M. tanarius* were significantly higher than that of *H. taiwanensis*.

**Table 4.** Means ± standard deviations for root traits of the three species studied and F-values for one-way ANOVA.

| Root Traits | *H. taiwanensis* | *M. tanarius* | *M. paniculatus* | ANOVA |
|---|---|---|---|---|
| RD (kg m$^{-3}$) | 1.07 ± 0.30[b] | 1.78 ± 0.17[a] | 1.94 ± 0.21[a] | 15.39[**] |
| RLD (km m$^{-3}$) | 0.73 ± 0.31[b] | 1.38 ± 0.26[a] | 1.19 ± 0.21[a] | 6.54[*] |
| RM (g m$^{-3}$) | 1067.4 ± 302.9[b] | 1782.4 ± 174.6[a] | 1939.8 ± 210.8[a] | 15.56[**] |
| RSA (cm$^2$) | 2995.3 ± 959.4[b] | 5391.4 ± 786.8[a] | 5277.4 ± 325.4[a] | 5.23[*] |
| RTD (g cm$^{-3}$) | 0.15 ± 0.03[b] | 0.29 ± 0.04[a] | 0.29 ± 0.02[a] | 30.56[***] |
| RV (cm$^3$) | 377.5 ± 76.8[a] | 332.5 ± 58.5[a] | 360.0 ± 46.9[a] | 0.54 ns |
| SRL (m g$^{-1}$) | 0.69 ± 0.26[a] | 0.77 ± 0.07[a] | 0.62 ± 0.13[a] | 0.76 ns |

RD, root density; RLD, root length density; RM, root mass; RSA, root surface area; RTD, root tissue density; RV, root volume; SRL, specific root length. In each row, means of species' parameter ($n = 14$) sharing different superscripts are significantly different from each other (Tukey's HSD post hoc test). Levels of significance are: [*]$p < 0.05$; [**]$p < 0.01$; [***]$p < 0.001$.

### 3.4. Root Anchorage Capability

Vertical pullout data showed pullout resistance force increased with displacement until it reached a peak and then decreased markedly as roots ruptured or slipped from the soil. Figure 3 shows typical vertical pullout force-displacement curves for the three species. The maximum pullout resistance force differed significantly among species and was highest for *M. paniculatus* (2.96 ± 0.58 kN) and lowest for *M. tanarius* (1.53 ± 0.64 kN) and *H. taiwanensis* (1.02 ± 0.33 kN) (Table 5). Regression analysis showed a positive relationship between maximum pullout resistance force and some root traits, that is, root collar diameter, root biomass and shoot biomass for all three species (Table 6). Linear regressions of

pullout resistance (Pr) and root collar diameter (RCD) for *H. taiwanensis*, *M. tanarius*, and *M. paniculatus* seedlings are: $Pr = 0.3215RCD - 4.3803$ ($R^2 = 0.7427$, $\alpha = 0.05$), $Pr = 0.3135RCD - 6.2987$ ($R^2 = 0.8413$, $\alpha = 0.01$) and $Pr = 0.2212RCD - 2.3782$ ($R^2 = 0.7559$, $\alpha = 0.05$), respectively. Linear regressions of pullout resistance (Pr) and root biomass (RB) for *H. taiwanensis*, *M. tanarius*, and *M. paniculatus* seedlings are: $Pr = 0.0236RB - 0.4553$ ($R^2 = 0.8276$, $\alpha = 0.01$), $Pr = 0.0715RB - 5.6277$ ($R^2 = 0.8270$, $\alpha = 0.01$) and $Pr = 0.0646RB - 3.8727$ ($R^2 = 0.8959$, $\alpha = 0.01$), respectively. Moreover, linear regressions of pullout resistance (Pr) and shoot biomass (SB) for *H. taiwanensis*, *M. tanarius*, and *M. paniculatus* seedlings are: $Pr = 0.0301SB - 0.8055$ ($R^2 = 0.7122$, $\alpha = 0.05$), $Pr = 0.0121SB - 0.6939$ ($R^2 = 0.6969$, $\alpha = 0.05$) and $Pr = 0.0179SB + 0.2498$ ($R^2 = 0.7251$, $\alpha = 0.05$), respectively (Table 6).

Multicollinearity diagnostic tests showed that the variance inflation factors (VIF) of root collar diameter and taproot length for *H. taiwanensis*, *M. tanarius* and *M. paniculatus* were 1.029, 1.025 and 1.065, respectively, indicating no collinearity between the predictor variables (root collar diameter and taproot length). The derived multiple linear regression equations are:

For *H. taiwanensis* seedlings, $Pr = 0.341RCD + 0.014TL - 5.33$ ($R^2 = 0.833^*$, r = 0.913, p = 0.028, $\alpha = 0.05$, VIF = 1.029).

For *M. tanarius* seedlings, $Pr = 0.32RCD + 0.006TL - 6.94$ ($R^2 = 0.855^*$, r = 0.925, p = 0.021, $\alpha = 0.05$, VIF = 1.025).

For *M. paniculatus* seedlings, $Pr = 0.246RCD - 0.009TL - 2.35$ ($R^2 = 0.907^{**}$, r = 0.952, p = 0.009, $\alpha = 0.01$, VIF = 1.065).

Where Pr is pullout resistance, RCD is root collar diameter, TL is taproot length, $\alpha$ is significance level (Table 7). Collectively, the root anchorage capability of *M. paniculatus* is higher than that of *M. tanarius* and *H. taiwanensis*.

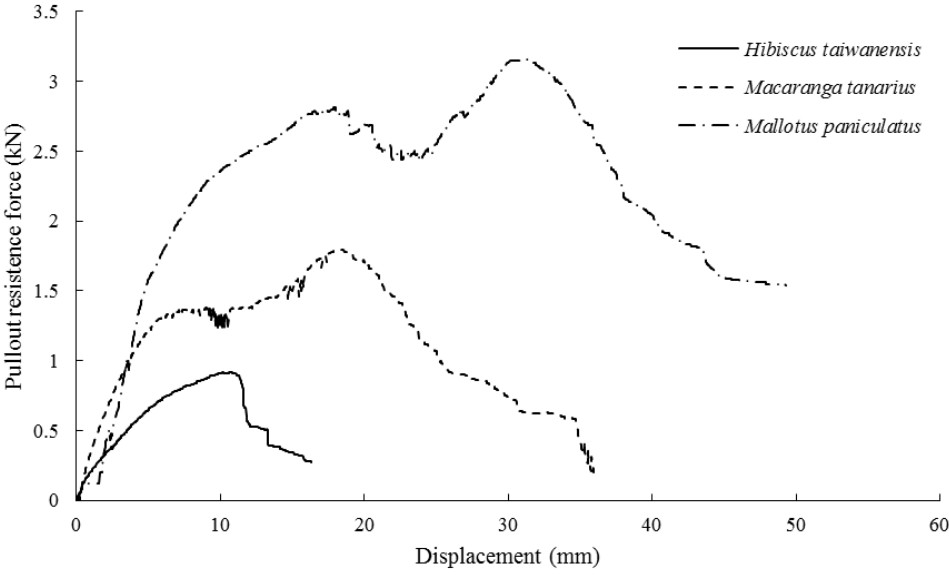

**Figure 3.** Typical pullout force-displacement curves for the three species studied.

**Table 5.** Means ± standard deviations of maximum pullout resistance force for the three species studied and the F-values for a one-way ANOVA.

| Biomechanical Properties | *H. taiwanensis* | *M. tanarius* | *M. paniculatus* | ANOVA |
|---|---|---|---|---|
| Maximum Pullout Resistance Force (kN) | 1.02±0.33[b] | 1.53±0.64[b] | 2.96±0.58[a] | 24.936[***] |

Different superscript letters in the same row indicate significant differences (Tukey's HSD post hoc test) among species. *N* = 14. Level of significance is: [***] *p* < 0.001.

**Table 6.** Relation between root traits and pullout resistance for the three species studied.

| Morphological traits | Species | Regression equation | $R^2$ | *p* |
|---|---|---|---|---|
| | *H. taiwanensis* | $P_r = 0.3215RCD - 4.3803$ | 0.7427* | 0.031 |
| RCD (mm) | *M. tanarius* | $P_r = 0.3135RCD - 6.2987$ | 0.8413** | 0.004 |
| | *M. paniculatus* | $P_r = 0.2212RCD - 2.3782$ | 0.7559* | 0.011 |
| | *H. taiwanensis* | $P_r = 0.007TL + 0.691$ | 0.023 | 0.746 |
| TL (cm) | *M. tanarius* | $P_r = -0.001TL + 1.641$ | 0.027 | 0.954 |
| | *M. paniculatus* | $P_r = -0.004TL + 3.216$ | 0.026 | 0.73 |
| | *H. taiwanensis* | $P_r = 0.00004064RT + 0.784$ | 0.054 | 0.617 |
| RT | *M. tanarius* | $P_r = 0.0001RT - 0.32$ | 0.17 | 0.358 |
| | *M. paniculatus* | $P_r = -0.00003811RT + 3.261$ | 0.01 | 0.835 |
| | *H. taiwanensis* | $P_r = 0.0236RB - 0.4553$ | 0.8276** | 0.004 |
| RB (g) | *M. tanarius* | $P_r = 0.0715RB - 5.6277$ | 0.8270** | 0.005 |
| | *M. paniculatus* | $P_r = 0.0646RB - 3.8727$ | 0.8959** | 0.001 |
| | *H. taiwanensis* | $P_r = 0.0301SB - 0.8055$ | 0.7122* | 0.017 |
| SB (g) | *M. tanarius* | $P_r = 0.0121SB - 0.6939$ | 0.6969* | 0.019 |
| | *M. paniculatus* | $P_r = 0.0179SB + 0.2498$ | 0.7251* | 0.015 |

$P_r$, pullout resistance; H, height; RCD, root collar diameter; TL, taproot length; RT, root tips; RB, root biomass; SB, shoot biomass. Levels of significance are: *$p < 0.05$; **$p < 0.01$.

**Table 7.** Relationships among pullout resistance, root collar diameter and taproot length for the three species studied.

| Species | Regression Equation | $R^2$ |
|---|---|---|
| *H. taiwanensi* | $Pr = 0.341RCD + 0.014TL - 5.33$ | 0.833* |
| *M. tanarius* | $Pr = 0.320RCD + 0.006TL - 6.94$ | 0.855* |
| *M. paniculatus* | $Pr = 0.246RCD - 0.009TL - 2.35$ | 0.907** |

$P_r$, pullout resistance; RCD, root collar diameter; TL, taproot length. Values are means ± standard errors of 14 replicates. Levels of significance are: *$p < 0.05$; **$p < 0.01$; ***$p < 0.001$.

## 3.5. Root Tensile Strength and Young's Modulus

The results showed great variation in tensile resistance and tensile strength among species. The highest mean tensile resistance force was observed with *M. paniculatus* (213.41 ± 46.98 N) and the lowest with *H. taiwanensis* (42.59 ± 7.46 N). The mean tensile strength of *M. paniculatus* (23.89 ± 8.93 MPa) was significantly higher than that of *M. tanarius* (17.77 ± 5.98 MPa) and *H. taiwanensis* (6.89 ± 3.71 MPa), respectively (Table 8). Furthermore, a significant difference was also observed in mean root tensile strength of different diameter classes between species (Table 9). In all root diameter classes, mean root tensile strength of *M. tanarius* and *M. paniculatus* were significantly higher than that of *H. taiwanensis*. Additionally, root tensile force increased with increasing root diameter following a power law function for the three species (Figure 4). However, root tensile strength decreased with increasing root diameter according to the power law (Figure 5).

**Table 8.** Means ± standard deviations of root diameter, root tensile resistance force and root tensile strength for the three species studied and F-values for one-way ANOVA.

| Parameters | *H. taiwanensis* | *M. tanarius* | *M. paniculatus* | ANOVA |
|---|---|---|---|---|
| Root diameter (mm) | 2.93 ± 1.83[a] | 3.47 ± 2.33[a] | 3.11 ± 2.41[a] | 0.448ns |
| Tensile resistance force (N) | 42.59 ± 7.46[b] | 197.04 ± 39.45[a] | 213.41 ± 46.98[a] | 6.612*** |
| Tensile strength (MPa) | 6.89 ± 3.71[c] | 17.77 ± 5.98[b] | 23.89 ± 8.93[a] | 50.764*** |

Different superscript letters in the same row indicate significant differences (ANOVA and Tukey's HSD post hoc test) among species. Levels of significance are: ns, non-significant; ***$p < 0.001$.

**Table 9.** Means ± standard deviations of root tensile strength of different root diameter classes for the three species studied.

| Root Diameter (mm) | Tensile Strength (MPa) | | |
|---|---|---|---|
| | *H. taiwanensis* | *M. tanarius* | *M. paniculatus* |
| 0-1 | 12.21 ± 4.5[b] | 26.78 ± 8.53[a] | 34.78 ± 10.01[a] |
| 1-2 | 8.13 ± 1.63[c] | 18.21 ± 4.57[b] | 27.06 ± 7.14[a] |
| 2-5 | 5.41 ± 1.89[b] | 16.98 ± 2.81[a] | 19.82 ± 2.04[a] |
| 5-10 | 3.97 ± 1.9[b] | 14.27 ± 4.18[a] | 17.37 ± 3.49[a] |

Different superscript letters in the same row indicate significant differences (ANOVA and Tukey's HSD post hoc test) among species. Level of significance is: $p < 0.001$.

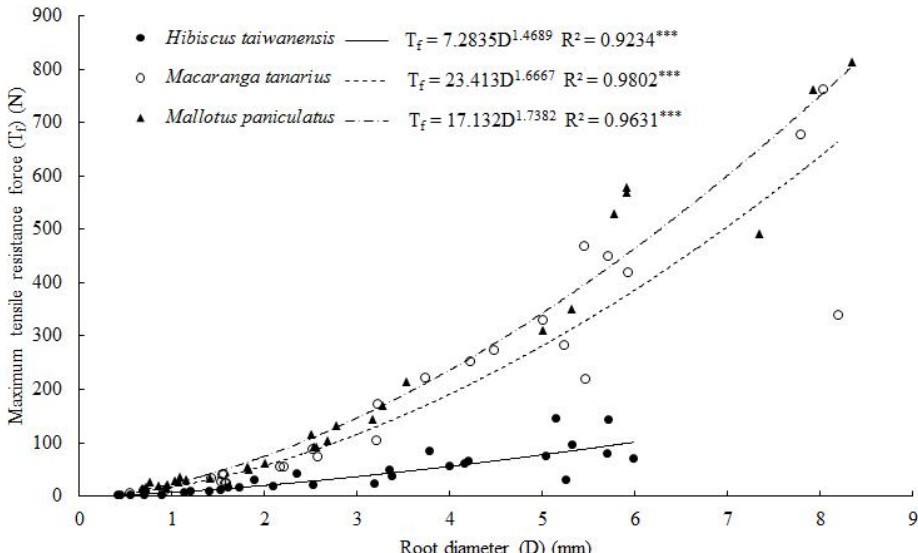

**Figure 4.** Relationship between root tensile resistance force and root diameter for the three studied species fitted to a power law function.

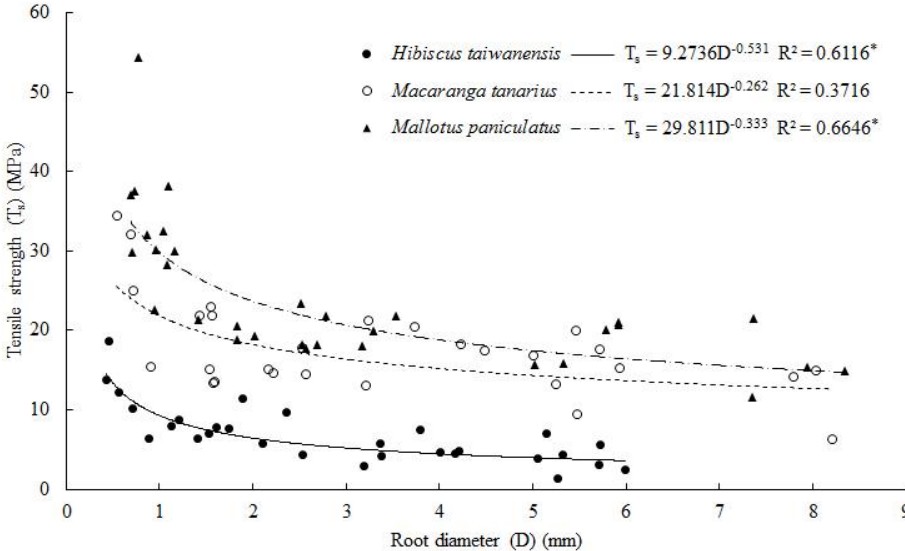

**Figure 5.** Relationship between root tensile strength and root diameter for the three studied species fitted to a power law function.

Results show the mean of Young's moduli in roots of *M. paniculatus* (127.15 ± 56.8 MPa) and *M. tanarius* (112.6 ± 64.9 MPa) were significantly higher than that of *H. taiwanensis* (25.27 ± 8.35 MPa)

(Table 10). Furthermore, significant differences were observed in Young's modulus in different diameter classes among species. Young's moduli of different root diameter classes for *M. tanarius* and *M. paniculatus* were significantly higher than that of *H. taiwanensis* (Table 11). Figure 6 shows that Young's modulus decreased with root diameter in accordance with a power law function. Moreover, Young's modulus increased with root tensile strength following a linear correlation for the three species (Figure 7).

**Table 10.** Means ± standard deviations of root Young's modulus for the three species studied and F-values for one-way ANOVA.

| Biomechanical Properties | *Hibiscus taiwanensis* | *Macaranga tanarius* | *Mallotus paniculatus* | ANOVA |
|---|---|---|---|---|
| Young's modulus (MPa) | 25.27 ± 8.35[b] | 112.6 ± 64.9[a] | 127.15 ± 56.8[a] | 26.346[***] |

Different superscript letters in the same row indicate significant differences (Tukey's HSD post hoc test) among species. Level of significance is: [***]$p < 0.001$.

**Table 11.** Means ± standard deviations of root's Young's modulus of different root diameter classes for the three species studied.

| Root Diameter (mm) | Young's Modulus (MPa) | | |
|---|---|---|---|
| | *H. taiwanensis* | *M. tanarius* | *M. paniculatus* |
| 0–1 | 78.44 ± 31.42[b] | 227.91 ± 71.72[a] | 205.55 ± 39.06[a] |
| 1–2 | 43.47 ± 20.25[b] | 109.31 ± 37.08[a] | 139.60 ± 38.11[a] |
| 2–5 | 23.60 ± 11.83[b] | 116.78 ± 33.59[a] | 109.06 ± 23.02[a] |
| 5–10 | 12.91 ± 5.41[b] | 59.35 ± 20.8[a] | 74.62 ± 27.33[a] |

Different superscript letters in the same row indicate significant differences (ANOVA and Tukey's HSD post hoc test) among species. Level of significance is: $p < 0.001$.

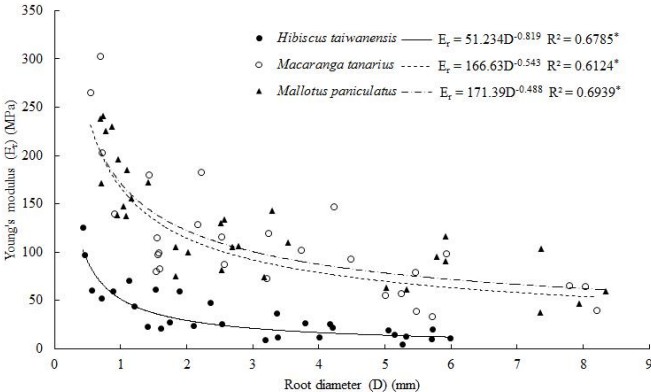

**Figure 6.** Relationship between Young's modulus and root diameter for the three species studied.

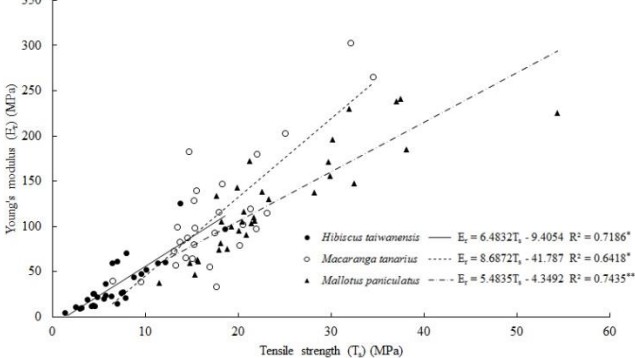

**Figure 7.** Relationship between Young's modulus and root tensile strength for the three species studied.

## 4. Discussion

### 4.1. Root System Architecture

This study shows that the root systems of *H. taiwanensis*, *M. tanarius* and *M. paniculatus* resemble VH- (vertical and horizontal) type, which is consistent with previous studies [16,19]. Several studies showed that trees with oblique roots will reinforce better the toe slopes, whereas trees with thick tap roots can strengthen soil better in the middle of the slope [40,41]. Fan and Chen [42] indicated that the VH-type root system is beneficial for slope stabilization and wind resistance, Safuddin and Normaniza [43] demonstrated that trees with VH-type root systems are recommended for planting in the middle of the slope. Thus, *H. taiwanensis* with shallow roots and a shorter taproot system is suggested for planting at the toe of the slop, while *M. tanarius* and *M. paniculatus* with deeper taproots and more profuse lateral root systems are recommended for planting in the middle of the slope. Our results also showed great variations in the root spatial distribution of *H. taiwanensis*, *M. tanarius* and *M. paniculatus* seedlings, and a sharp decrease in root density with increasing soil depth. Furthermore, RAR (root area ratio) values varied with soil depth and species. Most roots of the three species studied were spatially concentrated in the upper soil layer. In soil with poor fertility, the root system architecture plays an important role in resource acquisition [44,45]. Several studies have demonstrated that tree root density decreased with increasing soil depth [46,47]. Wasson et al. [48] indicated that plants with deeper roots and more lateral roots have a better ability to acquire nutrients and water. Thus, it is reasonable to assume that *M. tanarius* and *M. paniculatus* are more competitive than *H. taiwanensis*.

### 4.2. Seedling Growth Performance

Our results show that there are significant differences in seedling growth performance and root traits among species. Most traits, such as root collar diameter, taproot length, number of root tips, total root length and root biomass were significantly higher for *M. tanarius* and *M. paniculatus* seedlings than for *H. taiwanensis* seedlings. On the other hand, Table 3 showed that shoot height and shoot biomass were significantly higher for *M. paniculatus* seedlings than for *H. taiwanensis* and *M. tanarius* seedlings. The soil from landslide site used in this study had very low fertility as shown in Table 1. Generally, fatal landslide events were more frequent in the mountainous regions of Taiwan during monsoon and typhoon seasons. Landslides normally remove current vegetation and topsoil. Seedling growth performance of pioneer species are of vital importance in the rehabilitation of landslide areas [49]. Several studies showed that seedlings with larger root collar diameter have better survival and growth performance than seedlings with smaller root collar diameter [50,51]. Grossnickle [52] indicated that greater root mass equates to larger root absorptive surface and greater seedling drought tolerance capability. *H. taiwanensis*, *M. tanarius* and *M. paniculatus* are pioneer species playing vital roles in landslide reclamation and autogenic succession. Our findings demonstrate that *M. tanarius* and *M. paniculatus* seedlings with better growth performance than *H. taiwanensis* seedlings are more advantageous for soil bioengineering applications on landslide scars.

### 4.3. Root Traits

All root morphological traits, except for root volume and specific root length varied significantly among the three species studied. Root density, root length density, root mass, root surface area and root tissue density were significantly higher for *M. tanarius* and *M. paniculatus* seedlings than for *H. taiwanensis* seedlings. Previous studies indicated that root morphological traits (such as number of root tips, root surface area, root tissue density, and specific root length) have a great impact on the uptake of water and nutrients [53,54]. Kramer-Walter et al. [55] indicated that high root tissue density is the most predictable root trait that reflects adaptation to soils with low fertility. Burylo et al. [56] reported that root tissue density and total root surface area significantly affect plant anchorage. Root density and root length density are important parameters for evaluating erosion control effectiveness

of a species [57–59]. Moreover, our results show that total root length of *M. tanarius* and *M. paniculatus* seedlings is twice more than that of *H. taiwanensis* seedlings. Although root traits are affected by genetic constraints, it is highly plastic in response to various environmental stresses [60,61]. Generally, native pioneer tree species can survive and grow better in the harsh environmental conditions, and are ideal for successional reclamation of forest landslides. Taken as a whole, our findings demonstrate that *M. tanarius* and *M. paniculatus* seedlings with better root traits can adapt better to the challenging environment in the landslide scars.

### 4.4. Relationship between Root Morphological Traits and Anchorage Capability

The results of pullout testing showed that *M. paniculatus* seedlings have the highest maximum pullout resistance, which is 1.9 and 2.9 times higher than that of *M. tanarius* and *H. taiwanensis* seedlings, respectively. Correlation analyses of pullout resistance and morphological traits reveals a strong positive correlations with root collar diameter, root biomass and taproot length. These findings are consistent with the results of previous studies [56,62,63]. In general, tree roots play a vital role in shallow slope stabilization. Edmaier et al. [64] also found that the maximum uprooting force Avena sativa and Medicago sativa seedlings increases linearly with increasing total root length following a power law, and is mainly dependent on taproot length. Burylo et al. [58] also indicated that root system with a long tap root and numerous fine lateral roots are the best to resist erosion. Moreover, Yang et al. [65] demonstrated that the root collar diameter and taproot length are the key factors contributing to tree anchorage of *Pinus pinaster*. Leung et al. [66] showed that trees have higher pullout resistance than shrubs, and planted trees possess higher anchorage capability than natural ones. Collectively, *M. paniculatus* has the highest anchorage capability among the three species examined, and is ideal for ecological rehabilitation of landslides.

### 4.5. Root Tensile Resistance, Tensile Strength and Young's Modulus

In general, root tensile strength and Young's modulus play an important role in soil reinforcement [66–68]. Our results reveal significant differences in root tensile resistance, tensile strength and Young's modulus between species, which is similar to previous studies [36,69–71]. We also demonstrated that for the three species, root tensile resistance force increases with increasing root diameter following a positive law function, and root tensile strength decreases with increasing diameter in accordance with a negative power law function, which are consistent with earlier studies [21,23,57,69,72–74]. Genet et al. [69] also showed that tree root cellulose content increases with decreasing root diameter and increasing root tensile strength. On the other hand, Zhang et al. [75] attributed the relationship to the ratios of lignin/cellulose and lignin/alpha-cellulose decreased with increasing root diameter. The mean Young's moduli of roots of *M. paniculatus* and *M. tanarius* seedlings were about five-fold that of *H. taiwanensis* seedlings. Harhaway and Penny [76] reported that Young's modulus is positively correlated with cellulose content and negatively correlated with lignin/cellulose ratio. Furthermore, we found a strong linear relation between Young's modulus and tensile strength for *H. taiwanensis*, *M. tanarius* and *M. paniculatus*, respectively. The results of this study are congruent with previous studies [39,77]. Further research is needed to elucidate the effects of chemical constituents of roots on tensile strength and Young's modulus in these species.

In Taiwan, torrential rains accompanying tropical cyclones frequently trigger severe landslides in forestlands. Nowadays, reforestation has become a core issue of sustainable forest management. *H. taiwanensis*, *M. tanarius* and *M. paniculatus* are important pioneer tree species for reforestation on landslide scars. They play vital roles in soil reinforcement and land reclamation as well as vegetation succession. This study showed that there are significant differences in growth performance, root traits, anchorage capability, tensile strength and Young's modulus between species, suggesting that the priority order for species selection comes as: *M. paniculatus* > *M. tanarius* > *H. taiwanensis*. This study also demonstrates that root traits and biomechanical properties can be used as important factors for selecting potential tree species for reforestation of landslides. The processes naturally occurring after pioneer species establish show that the pioneer species can grow for about twenty years in landslide scars before they are gradually succeeded by shade-tolerant tree species. For efficient management,

we thus believe that the pioneer species could be thinned out about 20 years after establishment. Special silvicultural practice such as mixed planting with other native trees, grasses and shrubs may be implemented in order to enhance restoration of self-sustaining native forest ecosystems.

## 5. Conclusions

Our study highlights the significant differences in growth performance, root system architecture, root traits and biomechanical properties of three pioneer tree species. The root systems for *H. taiwanensis*, *M. tanarius* and *M. paniculatus* all resemble VH-type. However, *M. paniculatus* and *M. tanarius* seedlings develop deeper and more abundant root system than *H. taiwanensis* seedlings. *M. paniculatus* and *M. tanarius* seedlings also exhibit significantly higher growth performance than *H. taiwanensis* seedlings. In addition, root density, root length density, root mass, root surface area and root tissue density of *M. paniculatus* and *M. tanarius* seedlings are significantly higher than that of *H. taiwanensis* seedlings. The anchorage capacity and root tensile strength of *M. paniculatus* seedlings is significantly higher than that of *M. tanarius* and *H. taiwanensis* seedlings. Furthermore, Young's moduli of roots of *M. paniculatus* and *M. tanarius* seedlings are significantly higher than that of *H. taiwanensis* seedlings. Taken together, these results evidently suggest that the priority order for species selection rehabilitation of landslides is: *M. paniculatus* > *M. tanarius* > *H. taiwanensis*. Future studies on the management of these pioneer species for efficient restoration on landslide areas and the effect of selective thinning are needed.

**Author Contributions:** Conceptualization, J.-T.L.; Data curation, J.-T.L. and M.-Y.C.; Formal analysis, M.-Y.C., Y.-S.L.; Funding acquisition, J.-T.L.; Investigation, J.-T.L., M.-Y.C., Y.-S.L., K.-N.K. and W.-C.L.; Methodology, J.-T.L.; Project administration, J.-T.L. and M.-J.L.; Resources, M.-J.L., K.-N.K. and W.-C.L.; Software, J.-T.L., Y.-S.L. and M.-Y.C.; Supervision, J.-T.L.; Validation, J.-T.L.; Visualization, M.-J.L. Writing-original draft, J.-T.L. and M.-Y.C.; Writing-review and editing, J.-T.L., Y.-S.L. and M.-J.L. All authors have read and agreed to the published version of the manuscript.

**Funding:** Ministry of Science and Technology of Taiwan (project No. MOST 107-2311-B-415-001).

**Acknowledgments:** The authors would like to acknowledge the Ministry of Science and Technology of Taiwan for funding this project. The authors are also grateful to Maurice S. B. Ku from Department of Bio-agricultural Science, National Chiayi University for helpful discussions, comments on the manuscript, and valuable suggestions.

**Conflicts of Interest:** The authors declare no conflict of interest.

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
