# Peer review of "Root Traits and Biomechanical Properties of Three Tropical Pioneer Tree Species for Forest Restoration in Landslide Areas"

_forests, doi:10.3390/f11020179_

Round 1
Reviewer 1 Report
I would like a better overall general explanation of what processes naturally occur after pioneer species establish - and if they are they usually managed, or just left unattended? So is it implicit that you believe that management of species should take place? Enrichment planting/ selective removal? The last sentence of the abstract is weak! Is it just implying that more investigations are required? Furthermore I am not convinced that all of the terms are explained clearly to the non-specialist. "Soil reinforcement capability" for example? BUT I do like the investigation and believe it is useful research and has been well carried out.
Author Response
Response to Reviewer 1 Comments
forests-686041 Reviewer 1 Comments
Reviewer 1
I would like a better overall general explanation of what processes naturally occur after pioneer species establish - and if they are they usually managed, or just left unattended? So is it implicit that you believe that management of species should take place? Enrichment planting/ selective removal? The last sentence of the abstract is weak! Is it just implying that more investigations are required? Furthermore, I am not convinced that all of the terms are explained clearly to the non-specialist. "Soil reinforcement capability" for example? BUT I do like the investigation and believe it is useful research and has been well carried out.
Point 1: I would like a better overall general explanation of what processes naturally occur after pioneer species establish - and if they are they usually managed, or just left unattended? So is it implicit that you believe that management of species should take place? Enrichment planting/ selective removal? The last sentence of the abstract is weak!
Response 1: We greatly appreciate the reviewer’s comments. The processes naturally occur after pioneer species establish show that these pioneer species will grow up to forest for about twenty years in landslide scars. Then, they are gradually succeeded by shade-tolerant tree species, such as Cinnamomum micranthum, Machilus thunbergii, Castanopsis formosana, Castanopsis indica and Cyclobalanopsis glauca. They are just left unattended. So it is implicit that we believe that management of pioneer species with selective removal should take place at 20 years of age. This has been revised in the manuscript accordingly. The last sentence of the abstract is revised as “These results, along with the knowledge on vegetation dynamics following landslide, allow us to better evaluate the efficiency and selective removal management of pioneer species for the resilience and sustainability of landslide management”. (L33-L36)
The section of discussion is revised in the manuscript as “The processes naturally occur after pioneer species establish show that these pioneer species can grow up to forest for about twenty years in landslide scars. Then, they are gradually succeeded by shade-tolerant tree species, such as Cinnamomum micranthum, Machilus thunbergii, Castanopsis formosana, Castanopsis indica and Cyclobalanopsis glauca. These pioneer species are just left unattended. So it is implicit that we believe that management of these pioneer species with selective removal should take place at 20 years of age”. (L 438-L443).
Point 2: Is it just implying that more investigations are required?
Response 2: Thanks to the reviewer for the constructive comments. These have been revised in the manuscript as “Furthermore, future studies on the management of these pioneer species with selective removal at 20 years of age, and the implementation of these three species in silvicultural practices of forest restoration in landslide areas are needed”. (L459-L461)
Point 3: Furthermore, I am not convinced that all of the terms are explained clearly to the non-specialist. "Soil reinforcement capability" for example? BUT I do like the investigation and believe it is useful research and has been well carried out.
Response 3: Thanks to the reviewer for the constructive comments. The terms have been explained as clearly as possible. “Soil reinforcement capability” has been revised to “anchorage capability”. (L64)

Reviewer 2 Report
Excellent paper apart from small changes pointed in the annexed file. A revision of the paper's English is necessary due to the high number of small errors that although do not impede the understanding of the work displays a lack of same English rules.

Author Response
Response to Reviewer 2 Comments
forests-686041 Reviewer 2 Comments
Reviewer 2
Excellent paper apart from small changes pointed in the annexed file. A revision of the paper's English is necessary due to the high number of small errors that although do not impede the understanding of the work displays a lack of same English rules.
Point 1: I think that a notebook is not a measuring device. The methods for obtaining these measurements must be better characterized in order, for example to understand the significance of the results displayed in figure 3.
Response 1: We are grateful to the Reviewer’s constructive comments and suggestions for improving the manuscript. This has been revised in the manuscript accordingly. (The displacement was measured with a displacement transducer (TML DP-2000F, Tokyo Measuring Instruments Laboratory, Tokyo, Japan). (L132-L134)
Point 2: This information is not relevant. The results would be saved in any possible support without any relevance for the analytical phase of the work.
Response 2: Thanks for the thoughtful review. This has been revised in the manuscript accordingly. (The data of tensile force and displacement were acquired and compiled with an automatic data acquisition system (U-Soft DA-8, U-Soft Tech, Taipei, Taiwan).). (L148-L150)
Point 3: This paragraph should be in the results chapter and not in the methodological one.
Response 3: Thanks for the constructive comment. This paragraph has been moved to the result chapter. (For each species, 60 root segments were tested. Among the 60 roots tested, 29 root segments of H. taiwanensis seedlings fractured in the middle section, 28 root segments of M. tanarius seedlings fractured in the middle section, while 32 root segments of M. paniculatus seedlings fractured in the middle section.). (L294-L297)
Point 4: The first column (0-10) lack the variation values.
Response 4: Thanks for the thoughtful comment. The variation values are shown in the first column (0-10 cm) of Table 2. (L199-L200)
Point 5: The figure should only indicate points for each depth interval and the correspondent variation span.
Response 5: Thanks for the thoughtful review. Figure 2 mainly shows the distribution trend of root area ratio. The root area ratio at different soil depth interval and the correspondent variation span are shown in Table 2.
Point 6: Errors or variation? I think that there are no errors but only variation between the obtained results.
Response 6: Thanks for the constructive comment. This has been revised in the manuscript accordingly. (L223)
Point 7: Why typical pullout forces? are they the average results? Again the observed variation should be included in the graphic.
Response 7: Thanks for the thoughtful review. Figure 3 shows the typical pullout force-displacement curves for the three species studies. They are not the average results. The observed variations are shown in Table 5. (L282-L284)
Point 8: showed instead of exhibitd.
Response 8: Thanks for the comment. This has been revised in the manuscript accordingly. (showed). (L321)
Point 9: A revision of the paper's English is necessary due to the high number of small errors that although do not impede the understanding of the work displays a lack of same English rules.
Response 9: Thanks for the constructive comment. The paper’s English has been revised by Professor Maurice S. B. Ku, School of Biological Sciences, Washington State University. Pullman, WA 99164-4236, USA, and Professor Robert Haggerty, College of Agricultural and Life Sciences, University of Idaho, Moscow, ID 83844-233
